# Twin-To-Twin Transfusion Syndrome Donor and Recipient and Their Subsequent Cognitive Functioning in Late Childhood as Juvenile Athletes—A Case Study

**DOI:** 10.3390/ijerph18052545

**Published:** 2021-03-04

**Authors:** Ilona Bidzan-Bluma

**Affiliations:** Institute of Psychology, Faculty of Social Sciences, University of Gdansk, 80-309 Gdansk, Poland; ilona.bidzan-bluma@phdstud.ug.edu.pl

**Keywords:** perceptual skills, donor, recipient, TTTS, football, neurological damage

## Abstract

*Objective*: It is estimated that twin-to-twin transfusion syndrome (TTTS) occurs in 10–15% of monochorionic twin pregnancies. One of the fetuses takes on the role of donor and the other of recipient. The treatment administered involves serial amnioreduction and laser photocoagulation of the communicating blood vessels. After TTTS, children may have deficiencies in psychomotor functioning, in particular in cognitive functions, expressive language, and motor skills. Few scientific reports indicate that twins after TTTS do not demonstrate significant differences in tests which measure intellectual functioning. *Methods*: The cognitive functioning of twins in the late childhood period was compared using the following tools: an analysis of their medical history, an interview with their parents, and neuropsychological tests allowing the evaluation of their whole profile of cognitive functions. *Case Study*: Cognitive functioning in the late childhood period was analyzed in a pair of 11-year-old male twins (juvenile athletes), a donor and a recipient, who had developed TTTS syndrome in the prenatal period. *Results*: Comparison of the cognitive functioning profile of the donor and recipient revealed that children with a history of TTTS develop normally in terms of cognitive and motor functioning in late childhood. A comparative analysis of the donor and recipient was more favorable for the recipient, who had a higher level of general intelligence, visual–motor memory, and semantic fluency. *Conclusions*: The fact that both the donor and the recipient chose to pursue athletics suggests that gross motor skills are their strongest suit. Playing sports as a method of rehabilitation of cognitive function of children born prematurely after TTTS could contribute to the improvement of cognitive functioning.

## 1. Introduction

### 1.1. Twin-To-Twin Transfusion Syndrome (TTTS)

#### 1.1.1. Prevalence and Characteristics

It is estimated that TTTS (twin-to-twin transfusion syndrome) occurs in approximately 10–15% of monochorionic twin pregnancies (i.e., 1 in 2500 twin pregnancies in general) and is due to the sharing of anastomoses, joint blood vessels that are typical for single zygote twin pregnancies (i.e., “third placental circulation” [1,2,3]).

TTTS is characterized by uneven blood flow between fetuses which share a common placenta and have separate bags of amniotic fluid [4]. The resulting abnormal fetal blood volume and associated compensatory physiological responses, if not treated, lead to death, organ damage, and premature delivery [1,5].

#### 1.1.2. Donor and Recipient

One of the fetuses assumes the role of a donor and the other the recipient. The donor experiences symptoms associated with hypovolemia, hypotension, decreased diuresis, oligohydramnios, hypotrophy, and intrauterine fetal death. In contrast, the recipient is characterized by hypervolemia, hypertension, cardiomegaly, cardiomyopathy, circulatory failure, polyuria, polyhydramnios, and fetal edema, as well as intrauterine death [2,3].

#### 1.1.3. Therapeutic Interventions

When TTTS is diagnosed, it is treated with sequential selective laser photocoagulation of communicating vessels (SLPCV; closure of blood vessels located on the vascular “equator” of the placenta with a laser) [6,7,8]. SLPCV therapy may be used in pregnancies between 15 and 26 weeks. This method is associated with relatively low perinatal mortality, reaching up to 25%. Without treatment, the perinatal mortality rate of TTTS children ranges from 80 to 100%. The first center in Poland to perform SLPCV (and presently one of two), since July 2005, was the Department of Perinatology in the Obstetrics Clinic at the Medical University of Gdansk, with whom the author of the present study is currently cooperating.

### 1.2. Cognitive Functioning in Children after TTTS

Cognitive functions include: memory, such as semantic memory, short-term and long-term verbal and visual memory, attention (e.g., concentration, selectivity, divisibility), visual–spatial functions, and executive functions; complex cognitive processes include thinking (abstract and creative thinking), self-control, inhibition processes, planning and language concepts (e.g., understanding, reading, talking, and phonological and semantic fluency), and principles of neuropsychological assessment [9].

Cognitive functions are associated with acquiring new knowledge, reproducing it, learning language, and with executive functions related to emotional and behavioural self-regulation, which allow planning skills and consciously initiating and controlling one’s behaviour (i.e., inhibiting inappropriate reactions) [10,11,12]. Cognitive functioning may be impaired as a result of abnormal, pathological changes which occur during pregnancy, such as in the case of TTTS. Since only a few clinical centres perform fetoscopic laser photocoagulation of communicating vessels in the prenatal period, there is little research on this issue. Researchers indicate that the psychomotor functioning of TTTS children is reduced in the area of cognitive and motor skills [2], as well as in terms of language (including expressive language and mixed-dyslexia) [13,14]. Twin-to-twin transfusion syndrome is associated with a higher risk of prematurity and prematurity may be also associated with neurodevelopmental problems [15].

Intelligence levels depend on, inter alia, gestational age, Quintero stage of TTTS severity, and head circumference [2,16]. Few reports indicate that twins after TTTS do not show significant differences on intellectual functioning tests in childhood [16,17,18], but some researchers have also found difficulties with executive functioning and neuro-optometric deficiencies, which interfere with neurocognitive functioning [14]. Apart from early developmental support for premature children (and those affected by TTTS), clinical observations also emphasize the potential rehabilitation benefits of physical activity (e.g., playing football) in this group of children.

### 1.3. Cognitive Functioning in Children Practicing Football

The positive effects of sports on cognitive functioning is currently a topic of research [19,20]. Football (soccer) involves prospective motor control as well as movement coordination [21] and improves verbal and visual–spatial memory and planning [22,23]. In addition, new and difficult tasks that occur when playing football stimulate the prefrontal cortex, cerebellum, and basal ganglia by engaging executive functions such as planning and self-control [24]. Elite players are distinguished by better inhibition of some motor reactions, cognitive flexibility, ability to maintain an alert state, and, above all, better metacognition compared to adolescent and younger players [25,26]. Furthermore, executive functions may be improved by vigorous sport activities [27].

Therapeutic activities for children with reduced executive functioning include football training, which improves the diminished executive functions [28,29,30]. There is a complete lack of previous studies investigating the use of sports as a neurorehabilitation method, ameliorating the impaired executive functions of either TTTS children or prematurely born children.

### 1.4. Research Objectives

The aim of this study was to: (1) assess the cognitive functioning in late childhood of prematurely born twins who experienced TTTS prenatally; (2) investigate whether they are characterized by impairment in the cognitive functions whose impairment is characteristic of TTTS; (3) compare the cognitive functioning of the donor and of the recipient; and (4) assess whether playing football has improved certain areas of cognitive functioning in the boys.

This study is a part of a larger research project on the cognitive functioning of young athletes (10–12 years old) undertaking various sports activities (gymnastics, football) including bioelectrical brain function, as compared to their peers who do not do sports professionally. The study took place in the period between March 2020 and May 2021.

The research project was reviewed and approved positively by the Ethical Committee (decision no. 27/2020) at the [BLIND].

## 2. Methods

Case study was the chosen method used here. A case study is a research methodology typically used in social and life sciences, and often used in clinical psychology, neuropsychology, and rehabilitation psychology. Case studies usually apply many methods simultaneously in order to perform a psychological diagnosis that is as deep as possible. It is a comprehensive method of scientific research. The choice of performing a case study, as qualitative research, is related to the idiographic epistemological approach of the researcher, i.e., it is associated with the belief that studying every particular case is valuable, not just looking for general laws [31]. This also means that the conclusions drawn from the studied case cannot be generalized to other people. Case studies are used when researching something unique, special, or interesting (e.g., twins affected by TTTS) or for studying interventions (e.g., the use of sports in such cases) [31,32,33]. As emphasized by researchers [31] describing the steps undertaken when using a case study approach, this method of research allows the researcher to take a complex and broad topic or phenomenon and narrow it down into manageable research questions. By collecting qualitative or quantitative datasets about the phenomenon, the researcher gains a more in-depth insight into the phenomenon than could be obtained using only one type of data. Data in case studies are often, but not exclusively, qualitative in nature. The case study is linked to a theoretical framework [34].

In this case, where there are no other cases available for replication, the researcher can adopt the single-case design.

This study used purposeful sampling, where the goal is to find individuals or cases that provide insights into the specific situation under study regardless of the general population [35].

### 2.1. Research Tools Used

The following tools were used to assess cognitive functioning. The Cattell Culture Fair Intelligence Test–version 2 (CFT 20-R; Polish version by Stańczak) [36] was used to measure general intelligence. This test has a high internal consistency of the general score as well as of both test parts (split-half reliability > 0.80). The Diagnosis of Cognitive Functions Battery–PU1 [37] was used to assess the cognitive functions that play an important role in performance at school, such as attention, working memory, and executive functions. This test is reliable (Cronbach alpha value = 0.82). Bioelectrical activity was used to measure slow and fast brain waves via EEG Biofeedback [38]. The International Physical Activity Questionnaire IPAQ (modified version for testing children) was used [39] to evaluate the intensity of the participants’ physical and sport activity, and their sporting histories were analyzed using a survey about sport activity created for this study.

Moreover, analysis of medical history as well as a structured interview with the parents were performed.

### 2.2. Case Study

The participants were twins (11 years old; their mother’s first pregnancy), born at 37 weeks by Caesarean section. B, the donor, had a body weight of 2020 g and height of 51 cm, while K, the recipient, was 2140 g and 51 cm; both obtained 8 points on the Apgar scale. The boys’ parents were university graduates and were both 27 when the children were born. The presence of TTTS was discovered during the pregnancy and was treated with laser photocoagulation.

The interview with the parents showed that in the period of early childhood, the children’s development was average; they had some cognitive problems, especially regarding the ability to maintain attention and with inhibition.

Since the beginning of primary school, a natural type of physical activity that the children enjoyed, football, was used to improve their cognitive functioning. The boys play football 5–7 times a week for 60–120 min each time. They spend about 15 h a week on organized sports activities.

#### 2.2.1. Emotional Functioning of the Boys

In terms of emotional functioning, the boys are characterized by extroverted behaviour, high energy, and physical activity during classes. In this area, one difference is observed between the boys: K, the recipient, can manifest emotional outbursts when he does not understand something, while B, the donor, does not display such behavior (Figure 1).

#### 2.2.2. Physical Functioning

The boys have very high physical fitness and good motor coordination, as well as good posture, balance, flexibility, and physical endurance (Figure 1). 

#### 2.2.3. Cognitive Functioning

Table 1 shows the results of the comparison of slow and fast brain waves of the recipient-donor pair via EEG biofeedback.

Mean values recorded while performing the diagnosis (eyes open, and eyes open, and performing the “serial sevens” task). Theta/Beta ratio—corresponding to attention concentration, Theta/SMR ratio—corresponding to the experienced emotional tension. The K-recipient scored somewhat better in both areas.

The boys were compared with a boy, S, who also played football, was also a twin, but was not affected by TTTS prenatally, as well as with a boy, T, who was also born prematurely, but who was not a twin, and was not physically active outside of the compulsory physical education classes at school (all the results were included in Figure 2).

The qualitative analysis revealed certain differences. S scored higher than the twins TT TTTS on the ability to maintain attention and alertness (S: high score; twins after TTTS: average scores), but he scored lower on semantic fluency. T, a boy who was born prematurely and did not receive stimulation from physical activity, scored lower on the ability to maintain attention (average level), inhibition (low level), and short-term visual–motor memory and attention (low score, similar to that of one of the twins). There were no areas of cognitive functioning in which T was characterized by higher scores than either of the remaining boys (S and the twins after TTTS).

Moreover, the twins were compared to a pair of boys born at term (chosen randomly using a non-probabilistic method; i.e., random selection from a group of 30 children who play football, using systematic sampling).

This analysis revealed differences favoring the twins after TTTS in terms of phonological fluency (the twins scored average or a the higher end of average, in contrast to the other boys who scored low), semantic fluency (one of the twins scored high, the remaining boys scored average), ability to plan (the twins and one of the other boys scored high, one of the other boys scored average), or auditory working memory (the twins and one of the boys scored average, or at the upper border of the norm, one of the other boys scored low).

The twins after TTTS scored lower than the other boys in two areas: short-term visual–motor memory and attention (one of the twins scored average and the other low, while the other boys scored high), and on executive function monitoring (the twins and one of the boys scored average, while the other boy scored high).

## 3. Discussion

Children after TTTS syndrome are at high risk both in terms of developmental fluctuations [2,13,40,41,42], as well as in terms of pathologies in early environmental adaptation mechanisms [43,44,45,46].

The normal functioning of twins after TTTS syndrome in terms of cognitive and motor functions can be attributed to early tactile and vestibular stimulation in the prenatal period, which is consistent with studies indicating normal cognitive, motor, and emotional functioning [42,47,48,49,50,51,52]. The twins differ slightly in cognitive functioning and general intelligence: the functioning of K, the recipient, was at the upper limit of the average level, while that of B, the donor, was at the average level. The biggest difference concerns their visual–motor memory capacity, which was low in B and average in K. Both had difficulties performing two activities simultaneously, needed more time to analyze their errors, and had some difficulty quickly memorizing visual–spatial material. Additionally, they may both have had occasional difficulties formulating longer verbal statements and performing tasks with several stages. Most studies do not report differences in long-term impairment between donor and recipient twins [52]. However, Polish studies thus far have found better functioning in the recipient [2]. This can be influenced by both a more favorable prenatal environment and adequate prenatal care, which allows deficits in functioning to be minimized, even after the occurrence of such large differences in the initial prenatal stages. The introduction of appropriate incubators to hospitals (sound-proofed, dark, and with suitable mattresses [53,54]), ensuring constant contact with the parent, and kangaroo care (activities conducted at the hospital in which the children were born) have a significant impact on the subsequent development of the children, as has been confirmed in other studies [55,56,57]. Taking into account the decreased functioning of small children affected by TTTS, we did not observe difficulties in executive functioning in the boys (their levels of functioning in that area, depending on the function, were between average, i.e., similar to most peers, and high). The boys also had normal results on tests of verbal fluency which employ both language and executive functions. A comparison with prematurely born children (who do and do not play sports) was performed because many TTTS twin pregnancies do not go to term, and the long-term effects of TTTS are similar to the long-term effects of prematurity.

However, it needs to be stressed that in advanced cases of TTTS, infants may have long-term effects beyond the problems of prematurity. Intraventricular hemorrhage and other brain lesions are more common in TTTS babies, even after laser treatment or amnioreduction. If the disease is untreated and not followed closely, long-term effects may include heart failure and the death of one or both twins.

Playing sports could also contribute to the improvement of cognitive functioning [19,20]. Each sport has different characteristic features and requires different abilities from those who play it, and thus may influence different cognitive functions. In the case of football and its influence on the prefrontal cortex [24], and thus executive functioning, we also observed normal development in this domain in boys after TTTS. Earlier studies on football players have shown that they have better planning skills, and the hereby studied boys were indeed characterized by high levels of this skill [23].

## 4. Conclusions

Comparison of the donor and recipient in late childhood indicates that:−TTTS children show normal cognitive and motor development, which could have been influenced by both stimulation in early development and doing sports;−Slight differences are observed between the donor and recipient (in favor of the recipient) in terms of general intelligence and the scope of visual–motor memory;−The fact that the children choose to pursue athletics suggests that gross motor skills are their strongest suit;−Playing sports as a method of rehabilitation in cognitive function of children born prematurely after TTTS could contribute to the improvement of cognitive functioning.

### 4.1. Strengths

The unique aspect of this case study is the fact that twin-to-twin transfusion syndrome is a rare condition in twin pregnancies. For this reason, few articles exploring this topic and investigating the cognitive functioning of such twins have been published so far. Moreover, the authors are not aware of any paper on the cognitive functioning in late childhood of TTTS twins who are also athletes. The cases discussed in the available literature are usually characterized by cognitive function disorders. In addition, the research is usually quantitative, instead of qualitative, and is conducted on a small study group.

The presented neuropsychological research focuses on all areas of cognitive functioning (memory and attention processes as well as visual–spatial, executive, and language functions). The use of a case study allowed for an in-depth analysis of a single phenomenon. Unlike quantitative analysis, which observes patterns in data at the macro level on the basis of the frequency of the phenomena being observed, case studies observe data at the micro level [33]. Through case study methods, researchers can go beyond quantitative statistical results and understand behavioral conditions through the actor’s perspective. By including both quantitative and qualitative data, case studies help explain both the process and outcome of a phenomenon through complete observation, reconstruction, and analysis of the cases under investigation [33,34].

This research also has practical implications: playing sports as a method of rehabilitation of cognitive function of children born prematurely after TTTS could contribute to the improvement of cognitive functioning.

### 4.2. Study Limitations

The limitations of this study include the fact that it is not longitudinal and that only one case is described, rather than several. Case studies provide little basis for scientific generalization since they use a small number of subjects, particularly those with only one subject.

### 4.3. Future Directions

A longitudinal study on these boys and research on larger groups of twins affected by TTTS who do and do not play sports could provide further valuable insight into this subject.

## Figures and Tables

**Figure 1 ijerph-18-02545-f001:**
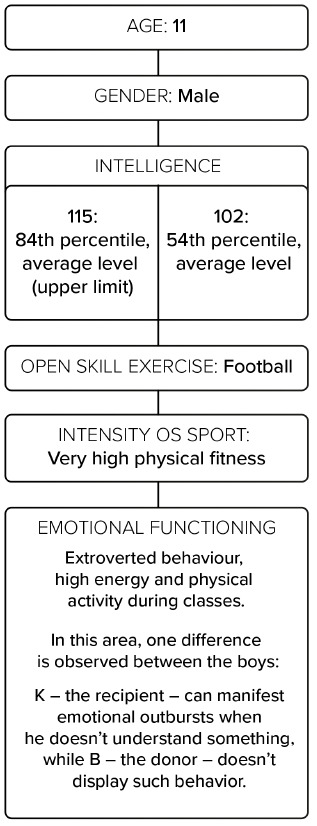
Author’s “hypothetical model” of the impact of individual elements by children after TTTS (twin-to-twin transfusion syndrome) on cognitive functions.

**Figure 2 ijerph-18-02545-f002:**
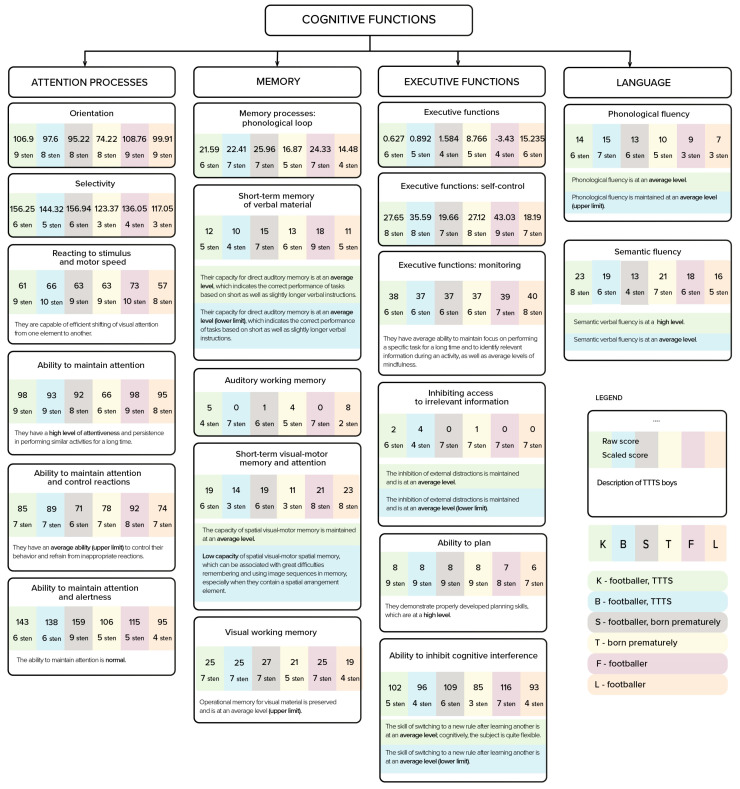
Profiles of cognitive functions.

**Table 1 ijerph-18-02545-t001:** Cognitive functioning and brain waves.

Tested Functions	K-Recipient	B-Donor
General intelligence, understood as liquid intelligence on the CFT 20-R	115: 84th percentile, average level (upper limit)	102: 54th percentile, average level
Theta/Beta ratio (related to attention when eyes are open)	3.38	3.19
Theta/SMR ratio (related to relaxation when eyes are open)	3.08	2.8
Theta/Beta ratio (related to attention when performing a task)	3.32	3.35
	Attention slightly improved during the task	Attention slightly worsened during the task
Theta/SMR ratio (associated with relaxation when performing a task)	3.13	3

## Data Availability

After the end of the project, the anonymized data will be attached to the project on OSF. For long-term data storage, the Open Science Framework Website (https://osf.io, accessed on 18 December 2020) will be used. Interview recordings will be stored on protected hard drive and university OneDrive cloud.

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
