# Peer review of "Twin-To-Twin Transfusion Syndrome Donor and Recipient and Their Subsequent Cognitive Functioning in Late Childhood as Juvenile Athletes—A Case Study"

_ijerph, 2021, doi:10.3390/ijerph18052545_

Round 1

Reviewer 1 Report

Thanks a lot for giving me the opportunity to review this paper titled "Twin-to-twin transfusion syndrome donor and recipient and their subsequent cognitive functioning in late childhood as juvenile athletes – a case study". This paper is very interesting and suitable with the remit and purpose of the journal even if some revisions needed to be solved before advice publication.

  • Although the introduction is well-written, the importance of the study still needs more clarification. Please, explain the rationale of the study.
  • Methods: Please define the duration of the study (start and end time).
  • What is the statistical method used for data analysis?
  • The discussion is very short and does not cover all the required elements. Please, frame it along the following lines:
  1. Main findings of the present study.
  2. Comparison with other studies.
  3. Implication and explanation of findings.
  4. Strengths and limitations.
  5. Conclusion, recommendation, and future direction.
  6. Explain in short about study limitations.
  7. Describe sources of potential bias and imprecision.
  8. Generalizability of the study findings needs to be put.
  9. The section needs to be as per well-defined objectives.
  10. It has to be framed in such a way that readers are able to have a good understanding of the current evidence and rationale of the paper.

Author Response

Dear Sir or Madame,

Thank you for reviewing our paper and sending your valuable comments and suggestions. Please find below our responses and comments.

Thanks a lot for giving me the opportunity to review this paper titled "Twin-to-twin transfusion syndrome donor and recipient and their subsequent cognitive functioning in late childhood as juvenile athletes – a case study". This paper is very interesting and suitable with the remit and purpose of the journal even if some revisions needed to be solved before advice publication.

  • Although the introduction is well-written, the importance of the study still needs more clarification. Please, explain the rationale of the study.

Thank you for this comment, I have thoroughly explained the rationale of the study.

  • Methods: Please define the duration of the study (start and end time).

I have added this information in the research methods section.

  • What is the statistical method used for data analysis?

Due to the rarity of the syndrome, never before described in the literature, a case study was used, which is based on an in-depth qualitative data analysis. According to the relevant experts, statistical methods cannot be used on such a small sample (two people).

  • The discussion is very short and does not cover all the required elements. Please, frame it along the following lines:

The discussion has been made more in-depth.

  • Main findings of the present study.

Main findings of this research:

  • TTTS children show normal cognitive and motor development, which could have been influenced by both stimulation in early development and doing sports;
  • slight differences are observed between the donor and recipient (in favour of the recipient) in terms of general intelligence and the scope of visual-motor memory;
  • the fact that the children choose to pursue athletics suggests that gross motor skills are their strongest suit;
  • sports, as a method for rehabilitation of cognitive function in children born prematurely after TTTS, could contribute to the improvement of cognitive functioning.
  • Comparison with other studies.

I have mainly referred to the results of studies on premature children and children with TTTS because there are no studies which have investigated the use ofsports for the neurorehabilitation of impaired functions of both children with TTTS and prematurely born children.

  • Implication and explanation of findings.

My research also has practical implications: playing sports could be an effective method of rehabilitation of cognitive functioning in children born prematurely after TTTS.

  • Strengths and limitations.

This has been done.

  • Conclusion, recommendation, and future direction.

This has been done.

  • Explain in short about study limitations.

The limitations of this study include the fact that it is not longitudinal and that only one case is described, rather than several. Case studies provide little basis for scientific generalisation since they use a small number of subjects, especially thosewith only one subject.

  • Describe sources of potential bias and imprecision.

The small group used may be a source of bias and imprecision. However, due to the rarity of the disorder, it is very difficult to create larger groups.

  • Generalizability of the study findings needs to be put.

When selecting case study as the methodology, we were aware that it does not allow one to make generalisations. However, it should be stressed that case study is a valuable method for collecting data and presenting a problem. It definitely provides new information about individuals and phenomena, which can allow deeper understanding – in this case, the cognitive functioning in late childhood of two male twins born preterm after TTTS, for whom sport (football) was used as a natural method of rehabilitation of cognitive functioning.

  • The section needs to be as per well-defined objectives.

It has been rewritten.

  • It has to be framed in such a way that readers are able to have a good understanding of the current evidence and rationale of the paper.

It has been rewritten.

We would like to you use this opportunity to thank you for your suggestions and revisions. We did our best to incorporate your recommendations to this manuscript. All suggestions have been included.

Best wishes!

Reviewer 2 Report

The introduction needs a "guiding thread" between the parts, something that connects the reasoning the aim of the manuscript.

Also, the discussion needs to be further explored. It is necessary that this part of the manuscript discusses with other studies about the area, or that the author tries to justify his findings (present hypotheses for his findings). I suggest paying special attention to that part.

Author Response

Dear Sir or Madame,

Thank you for reviewing our paper and sending your valuable comments and suggestions. Please find below our responses and comments.

1) The introduction needs a "guiding thread" between the parts, something that connects the reasoning the aim of the manuscript.

Thank you for this note, I have improved this by introducing a section that connects the two parts. 

2) Also, the discussion needs to be further explored. It is necessary that this part of the manuscript discusses with other studies about the area, or that the author tries to justify his findings (present hypotheses for his findings). I suggest paying special attention to that part.

I have made the discussion more extensive. However due to the complete lack of similar studies, I could only discuss the study in the context of previous studies that are related, but not closely. No hypotheses were made, as one of the characteristics of the chosen methodology (case study) is that no hypotheses are made.

We would like to you use this opportunity to thank you for your suggestions and revisions. We did our best to incorporate your recommendations to this manuscript. All suggestions have been included.
Best wishes!

Reviewer 3 Report

Twin-To-Twin Transfusion Syndrome Donor and Recipient and Their Subsequent Cognitive Functioning in Late Childhood as Juvenile Athletes – a Case Study

 Reference ijerph-1062342

Comments to Authors and Editor

General comments

Strengths:  Although a significant number of studies have been published in the field of neurodevelopmental outcomes, Twin-To-Twin transfusion syndrome to cognitive functioning in late childhood looks promising for me.

Weaknesses: However, I believe the manuscript is not currently ready to publish in the International Journal of Environmental Research and Public Health. The manuscript needs a major revision for publication. The major weakness of this manuscript is to justify the findings from a single case study. The end point outcome would be different if Author selects another twin to establish a decision of cognitive function in late childhood. From methodological point of view, the study looks cause-and-effect relationship. To find out the true effect (cognitive functioning), it is important to observe variation in the variable assumed to cause (transfusion syndrome) the change in the other variable(s)

 (confounders). In this circumstance, I am reluctant to accept the findings without designing a standard hypothesis driven approach with optimum number of sample size to achieve a targeted power of the study. The cognitive function varies with large number of confounders which should take into account in the model to justify the findings.

Specific comments:

I am not sure why Author suddenly staretd addressing the benefit of sporting activities, particularly soccer, for improving cognitive function. Although I am not a clinician, all sporting activities and event generic physical activities improve the neurodevelopmental outcomes and cognitive function.

Author addressed a series of tools to assess the cognitive functioning, for example, CFT 20-R, the Diagnosis of Cognitive Functions Battery, Bioelectrical activity, IPAQ. I would prefer to address the reliability and validated measures because the tools already validated previously.

A series a tested function has been addressed in Table 1. However, there is no clear definition of these functions. The audience would be benefitted to see the adequate information of these functions from lay point of view.

Author Response

Dear Sir or Madame,

Thank you for reviewing our paper and sending your valuable comments and suggestions. Please find below our responses and comments.

General comments

Strengths:  Although a significant number of studies have been published in the field of neurodevelopmental outcomes, Twin-To-Twin transfusion syndrome to cognitive functioning in late childhood looks promising for me.

Thank you

Weaknesses:

 However, I believe the manuscript is not currently ready to publish in the International Journal of Environmental Research and Public Health. The manuscript needs a major revision for publication. The major weakness of this manuscript is to justify the findings from a single case study.

Because of the uniqueness of the studied problem, and lack of previous, similar studies, I chose case study as the method of analysing the data. Because of how rare the phenomenon is, it was not possible to perform a larger study; however, I have designed such a study for the future.

The end point outcome would be different if Author selects another twin to establish a decision of cognitive function in late childhood.

Thank you for your comment. I decided to compare preterm children born after TTTS with other preterm born boys (a preterm twin boy, who also plays soccer,andasingletonpreterm boy, who does not play sports) as well as with two boys born on time.

From methodological point of view, the study looks cause-and-effect relationship. To find out the true effect (cognitive functioning), it is important to observe variation in the variable assumed to cause (transfusion syndrome) the change in the other variable(s)

I have included the data from the interview with parents and the medical history of the twin s that indicated lower cognitive functioning.

In this circumstance, I am reluctant to accept the findings without designing a standard hypothesis driven approach with optimum number of sample size to achieve a targeted power of the study.

The cognitive function varies with large number of confounders which should take into account in the model to justify the findings.

I have not made any hypotheses, as in the methodology I chose (case study) one does not make hypotheses. However, I have added a section where I describe this methodology in more detail:

Methods

Case study was the chosen method used here. Case study is a research methodology typically used in social and life sciences, and often used in clinical psychology, neuropsychology, and rehabilitation psychology. Case studies usually apply many methods simultaneously in order to perform a psychological diagnosis that is as deep as possible. It is a comprehensive method of scientific research. The choice of performing a case study, as qualitative research, is related to the idiographic epistemological approach of the researcher – that is to say it is associated with the belief that studying every particular case is valuable, not just looking for general laws (Heale, Twycross, 2018). This also means that the conclusions drawn from the studied case cannot be generalised to other people. Case studies are used when researching something unique, special, or interesting (e.g., twins affected by TTTS) or for studying interventions (e.g., the use of sports in such cases; Neale et al., 2006; Zainal, 2007; Heale & Twycross, 2018). As emphasised by researchers (Heale & Twycross, 2018) describing the steps undertaken when using a case study approach, this method of research allows the researcher to take a complex and broad topic or phenomenon and narrow it down into manageable research questions. By collecting qualitative or quantitative datasets about the phenomenon, the researcher gains a more in-depth insight into the phenomenon than could be obtained using only one type of data. Data in case studies are often, but not exclusively, qualitative in nature. The case study is linked to a theoretical framework (Tellis, 1997).

In this case, where there are no other cases available for replication, the researcher can adopt the single-case design.

This study used purposeful sampling, where the goal is to find individuals or cases that provide insights into the specific situation under study regardless of the general population (Schoch, 2020).

Specific comments:

I am not sure why Author suddenly staretd addressing the benefit of sporting activities, particularly soccer, for improving cognitive function. Although I am not a clinician, all sporting activities and event generic physical activities improve the neurodevelopmental outcomes and cognitive function.

These children were offered a natural method of neurorehabilitation(sport), consistent with their developmental needs (physical activity) in order to improve the more basiccognitive functions, in particular concentration and inhibition.

Author addressed a series of tools to assess the cognitive functioning, for example, CFT 20-R, the Diagnosis of Cognitive Functions Battery, Bioelectrical activity, IPAQ. I would prefer to address the reliability and validated measures because the tools already validated previously.

Thank you for your comment; all tools are reliable, I have added reliability indicators in the text.

A series a tested function has been addressed in Table 1. However, there is no clear definition of these functions. The audience would be benefitted to see the adequate information of these functions from lay point of view.

I added the following text:

Cognitive functions include: memory, such as semantic memory, short-term and long-term verbal and visual memory, attention (e.g., concentration, selectivity, divisibility), visual-spatial functions, and executive functions; complex cognitive processes include thinking (abstract and creative thinking), self-control, inhibition processes, planning and language concepts (e.g., understanding, reading, talking, and phonological and semantic fluency), and principles of neuropsychological assessment (Halligan et al., 2003).

We would like to you use this opportunity to thank you for your suggestions and revisions. We did our best to incorporate your recommendations to this manuscript. All suggestions have been included.

Best wishes!

Round 2

Reviewer 1 Report

All corrections done by the authors are amended. I have no further comments